# Sexual behaviour and incidence of sexually transmitted infections among men who have sex with men (MSM) using daily and event-driven pre-exposure prophylaxis (PrEP): Four-year follow-up of the Amsterdam PrEP (AMPrEP) demonstration project cohort

Mark A. M. van den Elshout[1]☺*, Eline S. Wijstma[1]☺, Anders Boyd[1,2,3,4], Vita W. Jongen[1,2], Liza Coyer[1], Peter L. Anderson[5], Udi Davidovich[1,6], Henry J. C. de Vries[1,3,7], Maria Prins[1,3,4], Maarten F. Schim van der Loeff[1,3,4‡], Elske Hoornenborg[1‡], on behalf of the Amsterdam PrEP Project team in the HIV Transmission Elimination AMsterdam Initiative (H-TEAM)¶

1 Department of Infectious Diseases, Public Health Service of Amsterdam, Amsterdam, the Netherlands, 2 Stichting hiv monitoring, Amsterdam, the Netherlands, 3 Amsterdam UMC location University of Amsterdam, Amsterdam Institute for Immunology and Infectious Diseases (AII), Amsterdam, the Netherlands, 4 Amsterdam UMC location University of Amsterdam, Department of Infectious Diseases, Amsterdam, the Netherlands, 5 Department of Pharmaceutical Sciences, University of Colorado Anschutz Medical Campus, Aurora, Colorado, United States of America, 6 Department of Social Psychology, University of Amsterdam, Amsterdam, the Netherlands, 7 Amsterdam UMC location University of Amsterdam, Department of Dermatology, Amsterdam, the Netherlands

☺ These authors contributed equally to this work.
‡ MFSvdL and EH also contributed equally to this work.
¶ Membership of the Amsterdam PrEP Project team in the HIV Transmission Elimination AMsterdam Initiative (H-TEAM) is provided in supporting information file "S1 Annex."
* mvdelshout@ggd.amsterdam.nl

## Abstract

### Background

An increasing number of countries are currently implementing or scaling-up HIV pre-exposure prophylaxis (PrEP) care. With the introduction of PrEP, there was apprehension that condom use would decline and sexually transmitted infections (STIs) would increase. To inform sexual health counselling and STI screening programmes, we aimed to study sexual behaviour and STI incidence among men who have sex with men (MSM) and transgender women who use long-term daily or event-driven PrEP.

### Methods and findings

The Amsterdam PrEP demonstration project (AMPrEP) was a prospective, closed cohort study, providing oral daily PrEP and event-driven PrEP to MSM and transgender women from 2015 to 2020. Participants could choose their PrEP regimen and could switch at each three-monthly visit. STI testing occurred at and, upon request, in-between 3-monthly study

**Data Availability Statement:** Data are available upon reasonable request. The AMPrEP data are owned by the Public Health Service of Amsterdam. Original data can be requested by submitting a study proposal to the steering committee of AMPrEP. The proposal format can be obtained from datamanagersoz@ggd.amsterdam.nl. Request for further information can also be submitted through the same email address. The AMPrEP steering committee verifies each proposal for compatibility with general objectives, ethical approval and informed consent forms of the AMPrEP study and potential overlap with ongoing studies. There are no restrictions to obtaining the data and all data requests will be processed in a similar way.

**Funding:** The AMPrEP study received funding as part of the H-TEAM initiative from ZonMw (grant number: 522002003), the National Institute for Public Health and the Environment (RIVM), GGD research funds and the H-TEAM. The study drug and an unrestricted research grant for AMPrEP was provided by Gilead Sciences. The H-TEAM initiative is supported by the Aidsfonds Netherlands (grant number: 2013169), Stichting Amsterdam-Dinner Foundation, Gilead Sciences Europe Ltd (grant number: PA-HIV-PREP-16-0024), Gilead Sciences (protocol numbers: CONL-276-4222,CO-US-276-1712), Janssen Pharmaceuticals (reference number: PHNL/JAN/0714/0005b/1912fde), M.A.C. AIDS Fund and ViiV Healthcare (PO numbers: 3000268822, 3000747780). The funders had no role in study design, data collection and analysis, decision to publish, or preparation of the manuscript.

**Competing interests:** PA has received personal fees from Gilead, Merck, Viiv, and research support from Gilead, paid to his institution. EH's institute recieved PrEP medication and unconditional research funding for AMPrEP from Gilead Sciences. MFSvdL has served on Advisory Boards of MSD and Novosanis; an investigator-initiated study is funded by GSK; all payments are made to his institution. AB has received speakers fees from Gilead Sciences, Inc.

**Abbreviations:** AMPrEP, Amsterdam PrEP demonstration project; a(I)RR, adjusted(incidence) rate ratio; AUDIT, Alcohol Use Disorders Identification Test; CAS, condomless anal sex; CI, confidence interval; DBS, dried blood spot; DUDIT, Drug Use Disorder Identification Test; GP, general practitioner; HCV, hepatitis C virus; IR, incidence rate; IRR, incidence rate ratio; IQR, interquartile range; MHI-5, Mental Health Inventory-5; MSM, men who have sex with men; NSSS, New Sexual Satisfaction Scale; PrEP, pre-exposure prophylaxis;

visits. We assessed changes in numbers of sex partners and condomless anal sex (CAS) acts with casual partners over time using negative binomial regression, adjusted for age. We assessed HIV incidence and changes in incidence rates (IRs) of any STI (i.e., chlamydia, gonorrhoea, or infectious syphilis) and individual STIs over time using Poisson regression, adjusted for age and testing frequency.

A total of 367 participants (365 MSM) commenced PrEP and were followed for a median 3.9 years (interquartile range [IQR] = 3.4–4.0). Median age was 40 years (IQR = 32–48), 315 participants (85.8%) self-declared ethnicity as white and 280 (76.3%) had a university or university of applied sciences degree. Overall median number of sex partners (past 3 months) was 13 (IQR = 6–26) and decreased per additional year on PrEP (adjusted rate ratio [aRR] = 0.86/year, 95% confidence interval [CI] = 0.83–0.88). Overall median number of CAS acts with casual partners (past 3 months) was 10 (IQR = 3–20.5) and also decreased (aRR = 0.92/year, 95% CI = 0.88–0.97). We diagnosed any STI in 1,092 consultations during 1,258 person years, resulting in an IR of 87/100 person years (95% CI = 82–92). IRs of any STI did not increase over time for daily PrEP or event-driven PrEP users. Two daily PrEP users, and no event-driven PrEP users, were diagnosed with HIV during their first year on PrEP. Study limitations include censoring follow-up due to COVID-19 measures and an underrepresentation of younger, non-white, practically educated, and transgender individuals.

## Conclusions

In this prospective cohort with a comparatively long follow-up period of 4 years, we observed very low HIV incidence and decreases in the numbers of casual sex partners and CAS acts over time. Although the STI incidence was high, it did not increase over time.

## Trial registration

The study was registered at the Netherlands Trial Register (NL5413) https://www.onderzoekmetmensen.nl/en/trial/22706

## Author summary

### Why was this study done?

- Oral pre-exposure prophylaxis (PrEP) is medication that almost 100% effectively prevents HIV when taken as prescribed: daily or before and after sex.

- Information on how people use PrEP, their sexual behaviour, and how often they acquire sexually transmitted infections (STI), can be used to tailor PrEP care to the needs of PrEP users.

- Since people can benefit from PrEP for several years (i.e., for as long as they are vulnerable to HIV), it is important to study the outcomes of PrEP use over longer periods of time.

PY, person years; RCT, randomised clinical trial; RR, relative ratio; STI, sexually transmitted infection; TDF/FTC, tenofovir disoproxil / emtricitabine; TFV-DP, tenofovir diphosphate.

## What did the researchers do and find?

- Between 2015 and 2020, 365 men who have sex with men and 2 transgender women who were at risk for HIV in the Netherlands were provided oral PrEP and followed up every 3 months.

- We examined how participants use PrEP (daily or before and after sex), participants' sexual behaviour, and how often they tested positive for STIs or HIV, and whether these outcomes changed over 4 years of PrEP use.

- Over 4 years, most participants used PrEP correctly, and only 2 acquired HIV. The numbers of sex partners and anal sex acts without a condom with casual partners decreased over time, and STIs occurred frequently, but did not increase over time.

## What do these findings mean?

- Our findings indicate that PrEP effectively prevents HIV over longer time periods and support structural implementation of PrEP with easy access.

- Our findings support regular counselling and STI testing as part of PrEP care.

- Young, non-white, practically educated, and transgender individuals were underrepresented in the study population; this should be considered when applying these findings to broader populations of PrEP users.

## Introduction

Since the publication of results from the iPrEX study in 2010 [1], data on oral pre-exposure prophylaxis (PrEP) use to prevent HIV have been accumulating through various randomised clinical trials (RCTs), demonstration studies, and implementation projects. PrEP has shown to be acceptable, safe, and highly effective in preventing HIV acquisition, provided adherence is good, especially among men who have sex with men (MSM) [2,3]. As such, PrEP plays a pivotal role in achieving the UNAIDS goal of zero new HIV infections [4,5], and is being rolled out in an increasing number of countries [6].

With the introduction of PrEP, there was apprehension that condom use would decline and sexually transmitted infections (STIs) would increase. In 2018, a systematic review of 17 PrEP cohort studies among MSM reported increases in condomless sex among PrEP users and increased STIs diagnoses, but the median follow-up time of included studies was only 6 months (range 3 to 18 months) [7]. One of the larger studies to date reported stable STI incidence, despite increased receptive condomless sex acts, with a median follow-up of 22 months [8]. To the best of our knowledge, there are no studies with longer follow-up time evaluating behavioural trends and STI incidence rates among MSM on PrEP. However, since many PrEP users are expected to use PrEP for several years, such information would be needed to inform policy makers and clinicians of current and future PrEP programmes. Therefore, we prospectively assessed sexual behaviour and incidence rates of HIV and other STIs, including hepatitis C virus (HCV), among MSM on PrEP for up to 4 years. We also assessed switching between the daily and event-driven regimen, PrEP discontinuation, and adherence to PrEP.

## Methods

### Study design

The Amsterdam PrEP demonstration project (AMPrEP) was an open-label demonstration study conducted between 3 August 2015 and 1 December 2020 that included MSM and transgender women. Participants were offered a free-of-charge oral coformulation of emtricitabine and tenofovir disoproxil 200/245 mg (TDF/FTC) to be used as daily or event-driven PrEP. The study design, aim, and procedures have been described previously [9], and analyses of the first 24 months of follow-up and HCV incidence have been published before [10,11]. Briefly, participants attended 3-monthly study visits at the Centre for Sexual Health of the Public Health Service of Amsterdam, the Netherlands. Eligible were MSM and transgender women without HIV, who were ≥18 years old and had, in the 6 months prior to screening, a substantial likelihood to acquire HIV sexually [9]. Switching between daily PrEP and event-driven PrEP was allowed at each 3-monthly study visit. All AMPrEP participants provided samples for HIV, HCV, and STI testing at each study visit. We requested participants to also provide blood for dried blood spots (DBSs) to measure adherence at the 3 or 6 and 12, 24, and 48 months study visits. We tested for HCV every 12 months until December 2016 and every 6 months thereafter [11,12].

### Measures

Sociodemographic, psychosocial, clinical, and behavioural characteristics were collected via questionnaires. Sociodemographics collected at inclusion in AMPrEP were age, gender identity, self-declared ethnicity, place of residency, education level [13], employment status, income level, living situation, relationship status, and sexual preference. Behavioural and clinical characteristics included history of condomless anal sex (CAS) and bacterial STIs in the 6 months prior to inclusion. Self-reported number of sex partners and anal sex acts, including partner type and condom use, were recorded three-monthly. Participants self-reported half-yearly whether they engaged in chemsex, defined as the use of γ-hydroxybutyrate/γ-butyrolactone, methamphetamine or mephedrone prior to or during sex.

Psychosocial determinants were measured yearly. Sexual compulsivity was measured using the sexual compulsivity scale [14], with a score ≥24 being indicative of a greater impact of sexual thoughts on daily functioning and of an inability to control sexual thoughts or behaviours [15]. Sexual satisfaction was measured using the New Sexual Satisfaction Scale (NSSS) on a scale from 20 to 100 [16]. Symptoms of depression or anxiety were assessed using the Mental Health Inventory-5 (MHI-5) score, where a score of <60 indicated symptoms of depression or anxiety [17]. The Alcohol Use Disorders Identification Test (AUDIT) [18] and Drug Use Disorder Identification Test (DUDIT) [19] questionnaires were used to assess problematic alcohol and drug use, respectively; scores ≥8 are interpreted as indicative of alcohol-related or drug-related problems [20].

### Outcomes

We assessed the number of sex partners, number of anal sex acts, and number of CAS acts with casual partners in the past 3 months at each study visit.

We assessed the number of diagnoses of chlamydia, gonorrhoea, infectious syphilis (stage 1, 2, and recent latent syphilis), HCV, and HIV. We calculated incidence rates (IRs) as the number of visits with a diagnosis (including repeat infections) divided by the person years (PY) of follow-up. Diagnoses were laboratory-confirmed infections from samples taken during study visits or additional visits at the Centre for Sexual Health during follow-up. We defined any STI as having one or more bacterial STIs (i.e., chlamydia, gonorrhoea, or infectious

syphilis) at a visit. We stratified chlamydia and gonorrhoea infections by anatomical site (i.e., anal, urogenital, or pharyngeal) and defined any anal STI as having anal chlamydia or anal gonorrhoea. In calculating PY for IRs of bacterial STI, we assumed that infection occurred at the date of positive test and follow-up time recontinued after infection. We defined incident HCV infections according to clinical practice guidelines [21] and distinguished between primary infections and reinfections. In calculating PY for IRs of HCV, we assumed that the infection occurred midway between the last negative and first positive test. Follow-up time stopped after infection and continued after confirmed sustained virologic response. In calculating PY for IRs of HIV, we assumed that infection occurred midway between the last negative and first positive test, and follow-up time stopped after infection.

We evaluated the number and rates of any switch between regimens as well as switch from daily PrEP to event-driven PrEP and vice versa. We also evaluated the number and rate of PrEP discontinuations, which were defined as one of the following: (a) a duration between study visits lasting at least 9 months without self-reporting continuing PrEP elsewhere during this period; (b) reporting not having taken PrEP for at least 3 months (regardless of visit attendance); (c) attending a formal study exit visit without self-reporting continuation of PrEP elsewhere; or (d) being lost to follow-up. Loss to follow-up was defined as not attending a study visit in the 9 months prior to 15 March 2020 (i.e., the start of the COVID-19 lockdown measures in the Netherlands), while not having completed the 48-month visit. Finally, we calculated the proportion of participants who still used PrEP at 48 months after enrolment among those who could have reached 48 months of follow-up before censoring.

We calculated median levels of intracellular tenofovir diphosphate (TFV-DP) in DBSs and corresponding IQRs among daily PrEP users and report the proportion of daily PrEP users with good adherence (TFV-DP $\geq$700 fmol/punch) [22]. We do not report these outcomes for event-driven PrEP users, since TFV-DP does not indicate prevention-effective adherence to event-driven PrEP [23].

## Laboratory methods

Laboratory methods were described previously [10,24]. For DBS analyses of intracellular TFV-DP, the 48 month samples were measured using a 50:50 methanol:water extraction. Results were divided by 1.138 in order to compare them with the previous 70:30 extractions.

## Statistical methods

We excluded participants without any follow-up study visits. Follow-up began at PrEP initiation (i.e., "baseline") and continued until 48 months of individual follow-up, last study visit, HIV diagnosis, or 15 March 2020, whichever occurred first. We excluded periods between PrEP discontinuation (as described above) and PrEP re-initiation from follow-up time. We presented analyses for both the overall study population and stratified on daily PrEP or event-driven PrEP. PrEP regimen was included as a time-updated variable.

To analyse changes in sexual behaviour, we report the median and interquartile ranges (IQR) of sexual behaviour outcomes for each study visit. We modelled the year-on-year change in mean sexual behaviour outcomes (excluding baseline visits) using a mixed-effects negative binomial regression model with a random intercept and random slope across individuals to account for between-individual variability at baseline and during follow-up, respectively, and robust standard errors to ensure variance corresponded to individuals. We report relative ratios (RRs) and corresponding 95% confidence intervals (CIs), and used a Wald $\chi^2$ test to test for changes over time. We provide unadjusted estimates and estimates adjusted for age at baseline. We initially modelled age as a categorical variable based on its nonlinear association with

sexual behaviour outcomes. Following peer-review, we modelled age as restricted cubic splines with 4 knots at the 5th, 35th, 65th, and 95th percentiles, to minimise loss of information.

We calculated STI IRs per 100PY of follow-up and corresponding 95% CI based on a Poisson distribution. To obtain insight into regimen choice, regimen switching, and STI incidence, we calculated the IR difference between daily PrEP and event-driven PrEP users among all participants and, after peer-review, additionally for the subset of participants who ever switched PrEP regimens. We report two-sided $p$-values for the IR difference. To analyse changes in bacterial STI incidence, we calculated STI IRs per 100PY for each 3-monthly follow-up period. Because we were interested in the STI incidence within yearly intervals, and there was lack of evidence that STI incidence varied jointly across 3-monthly intervals within a given year ($p = 0.22$), we modelled the change in STI IRs in years 2, 3, and 4 compared to the first year on PrEP. We used Poisson regression with a gamma-distributed frailty, and added a random intercept across individuals. We report incidence rate ratios (IRRs) and corresponding 95% CI per year, and used a Wald $\chi^2$ test to test for changes compared to the first year on PrEP. We provide unadjusted estimates and estimates adjusted for age at baseline and individual yearly STI testing frequency (time-updated). We initially modelled age and STI testing frequency as categorical variables based on their nonlinear association with any STI incidence. Following peer-review, we modelled both variables as restricted cubic splines with 4 knots at the 5th, 35th, 65th, and 95th percentiles, to minimise loss of information. We did not model change in HIV incidence rates over time owing to the low number of infections.

To analyse changes in PrEP use, we calculated regimen switch rates per 100PY and 95% CI based on a Poisson distribution, and calculated linear change in switch rates per year as switch rate ratio. We calculated the total number of PrEP discontinuations (including discontinuations after re-initiating) and median time until first discontinuing PrEP. We evaluated factors associated with time until first stopping PrEP using multivariable Cox regression. The factors were selected a priori [25,26] and related to: sociodemographics (age, education level, and place of residence), sexual behaviour and STI (number of CAS acts with casual partners, any bacterial STI in the past 3 months), and mental wellbeing. We included education level and place of residency as time-fixed variables and all other variables as time-updated. Data from (half-)yearly questionnaires were carried backwards for study visits that occurred in the past 6 and 12 months, respectively.

In sensitivity analyses, we re-ran the models on STI incidence and sexual behaviour using follow-up time including periods between PrEP discontinuation and re-initiation as periods with missing data. Additionally, we re-ran the models on STI incidence and sexual behaviour among participants who never switched PrEP regimens.

We defined significance at a $p$-value <0.05. All statistical analyses were performed in STATA version 17.0 (StataCorp, College Station, Texas, United States of America). The study was registered at the Netherlands Trial Register (NL5413). This study is reported as per the Strengthening the Reporting of Observational Studies in Epidemiology (STROBE) guideline (S1 STROBE checklist).

## Role of the funders

The study funders had no role in study design, data collection, data analysis, data interpretation, nor in writing of the manuscript.

### Ethics approval

The study was approved by the ethics board of the Amsterdam University Medical Centers, location AMC, Amsterdam, the Netherlands (NL49504.018.14). Participants gave written informed consent to participate in the study before taking part.

## Results

### Participant characteristics and duration of follow-up

Between 3 August 2015 and 31 May 2016, 376 participants were enrolled. Of these, 9 (2.4%) were excluded from analyses because they did not attend any follow-up visits. Of the 367 included participants, 365 were MSM and 2 identified as transgender women. Median age at baseline was 40 years (IQR = 32–48), 315/367 (85.8%) self-declared to be white and 280/367 (76.3%) had a university/university of applied sciences degree (Table 1). The median follow-up time was 3.9 years (IQR = 3.4–4.0), totalling 1258PY of observation. Of 282 participants who could have reached 48 months of follow-up before censoring, 192 (68%) still used PrEP after 48 months.

### Sexual behaviour

Overall median number of sex partners (past 3 months) was 13 (IQR = 6–26), overall median number of anal sex acts was 18 (IQR = 9–34), and overall median number of CAS acts with casual partners was 10 (IQR = 3–20.5) (S1 Table). Numbers of sex partners and anal sex acts decreased with each additional year on PrEP, adjusted for age (adjusted rate ratio [aRR] 0.86/year [95% CI 0.83–0.88] and 0.88/year [95% CI 0.85–0.91], respectively; Table 2). These changes were also statistically significant when stratified by PrEP regimen (Table 2 and Fig 1). Number of CAS acts with casual partners in the past 3 months also decreased with each additional year on PrEP, when adjusted for age at baseline (aRR 0.92/year [95% CI 0.88–0.97]), also when stratified by regimen (Table 2 and Fig 1). Numbers of sex partners, anal sex acts, and CAS acts with casual partners were higher in daily PrEP users compared to event-driven PrEP users (S1 Table).

Results from sensitivity analyses assessing behaviour over time since initiating PrEP while including periods without PrEP use or follow-up were largely the same (S2 Table).

### Incidence of bacterial sexually transmitted infections

We diagnosed one or more bacterial STIs in 1,092 consultations among 289/367 participants, during 1258PY: 891 during 914PY among daily PrEP users and 201 during 344PY among event-driven PrEP users (Table 3). IR of any STI was 87/100PY (95% CI = 82–92). This was higher for daily PrEP users (97/100PY; 95% CI = 91–104) compared to event-driven PrEP users (59/100PY; 95% CI = 51–67; $p < 0.0001$; Fig 2 and Tables 3 and S3). Results were similar when the analysis was restricted to PrEP users who ever switched PrEP regimens (S4 Table). Compared to the first year, IRs of any STI were lower in the second (aIRR = 0.77, 95% CI = 0.65–0.91) and third (aIRR = 0.78, 95% CI = 0.66–0.92) years, also for chlamydia and gonorrhoea (Fig 3 and S5 Table), adjusted for age and STI testing frequency. This decrease was not seen in the fourth year (aIRR = 0.89, 95% CI = 0.75–1.06). When stratified by regimen, similar findings were observed among daily PrEP users, but IRs were stable over yearly intervals among event-driven PrEP users (Fig 3 and S5 Table). Sensitivity analyses assessing STI incidence since initiating PrEP and ignoring gaps in follow-up (S6 Table) and assessing STI incidence among participants who never switched PrEP regimens (S7 Table), yielded comparable results.

**Table 1. Baseline characteristics of 367 AMPrEP participants with follow-up, overall and by PrEP regimen chosen at baseline, Amsterdam, the Netherlands, 2015–2018.**

| | Total (N = 367) | Event-driven PrEP (n = 98) | Daily PrEP (n = 269) | p-value[a] |
|---|---|---|---|---|
| **Baseline characteristic** | N (%) | n (%) | n (%) | |
| **Age (years), median [IQR]** | 40 [32–48] | 44 [35–52] | 38 [30–47] | **<0.001** |
| ≤34 | 121 (33.0) | 23 (23.5) | 98 (36.4) | **0.016** |
| 35–44 | 111 (30.3) | 28 (28.6) | 83 (30.9) | |
| ≥45 | 135 (36.8) | 47 (48.0) | 88 (32.7) | |
| **Gender** | | | | |
| Male | 365 (99.5) | 97 (99.0) | 268 (99.6) | 0.463 |
| Transgender female | 2 (0.5) | 1 (1.0) | 1 (0.4) | |
| **Self-declared ethnicity** | | | | |
| White | 315 (85.8) | 87 (88.8) | 228 (84.8) | 0.329 |
| Non-white | 52 (14.2) | 11 (11.2) | 41 (15.2) | |
| **Residence** | | | | |
| Amsterdam | 223 (60.8) | 64 (65.3) | 159 (59.1) | 0.282 |
| Other | 144 (39.2) | 34 (34.7) | 110 (40.9) | |
| **Education level** | | | | |
| No university/university of applied sciences degree | 87 (23.7) | 16 (16.3) | 71 (26.4) | **0.045** |
| University/university of applied sciences degree | 280 (76.3) | 82 (83.7) | 198 (73.6) | |
| **Employed** | | | | |
| Yes | 283 (78.0) | 75 (76.5) | 208 (78.5) | 0.216 |
| No | 18 (5.0) | 8 (8.2) | 10 (3.8) | |
| Other (retired, volunteer, disabled, student) | 62 (17.1) | 15 (15.3) | 47 (17.7) | |
| **Monthly net income level (n = 16 missing)** | | | | |
| ≤€1,700 | 97 (27.6) | 25 (25.8) | 72 (28.4) | 0.538 |
| €1,701–€2,950 | 150 (42.7) | 39 (40.2) | 111 (43.7) | |
| >€2,950 | 104 (29.6) | 33 (34.0) | 71 (28.0) | |
| **In steady relationship (n = 4 missing)** | | | | |
| No | 204 (56.2) | 46 (47.4) | 158 (59.4) | **0.042** |
| Yes | 159 (43.8) | 51 (52.6) | 108 (40.6) | |
| **Living situation** | | | | |
| Alone | 195 (53.1) | 57 (58.2) | 138 (51.3) | **0.040** |
| With partner | 117 (31.9) | 34 (34.7) | 83 (30.9) | |
| With parents/flatmates | 55 (15.0) | 7 (7.1) | 48 (17.8) | |
| **Sexual preference (n = 1 missing)** | | | | |
| Exclusively homosexual | 289 (79.0) | 77 (79.4) | 212 (78.8) | 0.906 |
| Not exclusively homosexual | 77 (21.0) | 20 (20.6) | 57 (21.2) | |
| **Bacterial STI diagnosed at PrEP initiation** | | | | |
| Any STI (n = 2 missing) | 72 (19.7) | 16 (16.3) | 56 (21.0) | 0.323 |
| Any anal chlamydia or gonorrhoea (n = 12 missing) | 45 (12.7) | 9 (9.6) | 36 (13.8) | 0.292 |
| Any chlamydia (n = 9 missing) | 36 (10.1) | 6 (6.2) | 30 (11.5) | 0.138 |
| Anal chlamydia (n = 13 missing) | 24 (6.8) | 4 (4.3) | 20 (7.7) | 0.341 |
| Urogenital chlamydia (n = 11 missing) | 14 (3.9) | 1 (1.0) | 13 (5.0) | 0.124 |
| Pharyngeal chlamydia (n = 13 missing) | 8 (2.3) | 1 (1.0) | 7 (2.7) | 0.454 |
| Any gonorrhoea (n = 8 missing) | 35 (9.8) | 9 (9.3) | 26 (9.9) | 0.855 |
| Anal gonorrhoea (n = 12 missing) | 24 (6.8) | 5 (5.3) | 19 (7.3) | 0.516 |
| Urogenital gonorrhoea (n = 11 missing) | 3 (0.8) | 1 (1.0) | 2 (0.8) | 1.000 |
| Pharyngeal gonorrhoea (n = 12 missing) | 18 (5.1) | 5 (5.2) | 13 (5.0) | 0.965 |

*(Continued)*

**Table 1.** (Continued)

| | Total (N = 367) | Event-driven PrEP (n = 98) | Daily PrEP (n = 269) | p-value[a] |
|---|---|---|---|---|
| Syphilis (stage 1 or 2) (n = 9 missing) | 5 (1.4) | 1 (1.0) | 4 (1.5) | 1.000 |
| **Number of anal sex partners (past 3 months), median [IQR]** | 12 [6–25] | 9.5 [5–20] | 13 [7–25] | 0.008 |
| **Number of anal sex acts (past 3 months), median [IQR]** | 22 [11–36] | 20.5 [8–35] | 22 [12–37] | 0.109 |
| **Number of CAS acts (past 3 months), median [IQR]** | 11 [4–23] | 8 [3–18] | 12 [4–23] | 0.070 |
| **Number of CAS acts with casual partners (past 3 months), median [IQR]** | 6 [2–14] | 4 [1–9] | 7 [3–15] | **0.0007** |
| **Position during CAS (past 3 months)** | | | | |
| None | 26 (7.1) | 11 (11.2) | 15 (5.6) | 0.264 |
| Top only | 62 (16.9) | 15 (15.3) | 47 (17.5) | |
| Bottom only | 62 (16.9) | 18 (18.4) | 44 (16.4) | |
| Versatile | 217 (59.1) | 54 (55.1) | 163 (60.6) | |
| **Chemsex (past 3 months) (n = 5 missing)** | 155 (42.8) | 43 (44.8) | 112 (42.1) | 0.648 |
| **AUDIT (n = 4 missing)** | | | | |
| Score <8 (not indicative of problematic alcohol use) | 263 (72.5) | 73 (75.3) | 190 (71.4) | 0.470 |
| Score ≥8 (indicative of problematic alcohol use) | 100 (27.6) | 24 (24.7) | 76 (28.9) | |
| **DUDIT (n = 2 missing)** | | | | |
| Score <8 (not indicative of problematic drug use) | 230 (63.0) | 67 (69.1) | 163 (60.8) | 0.149 |
| Score ≥8 (indicative of problematic drug use) | 135 (37.0) | 30 (30.9) | 105 (39.2) | |
| **Depressive or anxiety symptoms** | | | | |
| MHI-5 score ≥60 (no symptoms) | 291 (79.2) | 76 (77.6) | 215 (79.9) | 0.619 |
| MHI-5 score <60 (symptoms) | 76 (20.7) | 22 (22.5) | 54 (20.1) | |
| **Sexual Satisfaction Scale** | 45 (39–48) | 45 (39–49) | 44 (37–48) | 0.111 |
| **Sexual Compulsivity Scale (n = 1 missing)** | | | | |
| Score <24 (not indicative of compulsive sexual behaviour) | 283 (77.3) | 79 (80.6) | 204 (76.1) | 0.363 |
| Score ≥24 (indicative of compulsive sexual behaviour) | 83 (22.7) | 19 (19.4) | 64 (23.9) | |

[a] Two-sided p-values were calculated using Wilcoxon rank sum test for continuous variables and Pearson's $\chi^2$ test or Fisher's exact test for categorical variables.

AMPrEP, Amsterdam PrEP demonstration project; AUDIT, Alcohol Use Disorder Identification Test; DUDIT, Drug Use Disorder Identification Test; IQR, interquartile range; MHI-5, 5-item Mental Health Inventory; PrEP, pre-exposure prophylaxis; STI, sexually transmitted infection.

## Incidence rates of HIV and HCV

Two daily PrEP users were diagnosed with HIV during follow-up, both in the first year since PrEP initiation, resulting in an IR of 0.2/100PY (95% CI = 0.0–0.6; Table 3). We have already reported on these cases in more detail [10,27]. In brief, one participant had discontinued PrEP several months before HIV diagnosis and the other participant was a daily PrEP user whose adherence to PrEP was confirmed to be adequate [27]. No diagnosis of HIV was observed among event-driven PrEP users (IR = 0.0/100PY, 95% CI = 0.0–1.1; Table 3).

The 17 incident HCV infections were diagnosed among 15 participants during 1186PY of follow-up (IR = 1.4/100PY, 95% CI = 0.9–2.3; Table 3). Ten were primary infections and 7 reinfections (S3 Table). Daily PrEP users accounted for 15 of these infections (IR = 1.7/100PY, 95% CI = 1.0–2.9) and event-driven PrEP users for 2 (IR = 0.6/100PY, 95% CI = 0.2–2.5; Table 3). IRs of HCV decreased over time: year 1, n = 6 (IR = 1.8/100PY, 95% CI = 0.8–3.9); year 2, n = 8 (IR = 2.5/100PY, 95% CI = 1.3–5.2); year 3, n = 2 (IR = 0.7/100PY, 95% CI = 0.2–2.7); year 4, n = 1 (IR = 0.4/100PY, 95% CI = 0.1–2.6; S5 Table).

**Table 2. Four-year outcomes of sexual behaviour per year on PrEP among 367 AMPrEP participants, Amsterdam, the Netherlands, 2015–2020.**

| | | Crude | | | Adjusted[b] | | |
|---|---|---|---|---|---|---|---|
| | No. of visits with data | RR (95% CI) | | p-Value | aRR (95% CI) | | p-Value[c] |
| **Number of sexual partners[a]** | | | | | | | |
| Overall | 5,105 | 0.89 | [0.86–0.91] | <0.0001 | 0.86 | [0.83–0.88] | <0.0001 |
| Daily | 3,787 | 0.91 | [0.88–0.94] | <0.0001 | 0.87 | [0.84–0.91] | <0.0001 |
| Event-driven | 1,318 | 0.85 | [0.81–0.90] | <0.0001 | 0.82 | [0.77–0.88] | <0.0001 |
| **Number of anal sex acts[a]** | | | | | | | |
| Overall | 5,112 | 0.91 | [0.88–0.93] | <0.0001 | 0.88 | [0.85–0.91] | <0.0001 |
| Daily | 3,791 | 0.93 | [0.90–0.96] | <0.0001 | 0.90 | [0.87–0.93] | <0.0001 |
| Event-driven | 1,321 | 0.85 | [0.80–0.90] | <0.0001 | 0.84 | [0.79–0.89] | <0.0001 |
| **Number of CAS acts with casual partners[a]** | | | | | | | |
| Overall | 5,108 | 0.97 | [0.94–1.01] | 0.17 | 0.92 | [0.88–0.97] | 0.0006 |
| Daily | 3,787 | 1.00 | [0.96–1.04] | 0.96 | 0.94 | [0.89–0.99] | 0.018 |
| Event-driven | 1,321 | 0.93 | [0.87–1.00] | 0.067 | 0.90 | [0.82–0.98] | 0.013 |

[a]In the past 3 months.

[b]Adjusted for age at enrolment modelled as restricted cubic splines with 4 knots, and including a random intercept and random slope at the participant-level.

[c]p-Value based on the Wald test.

AMPrEP, Amsterdam PrEP demonstration project; CAS, condomless anal sex; CI, confidence interval; PrEP, pre-exposure prophylaxis; (a)RR, (adjusted) rate ratio.

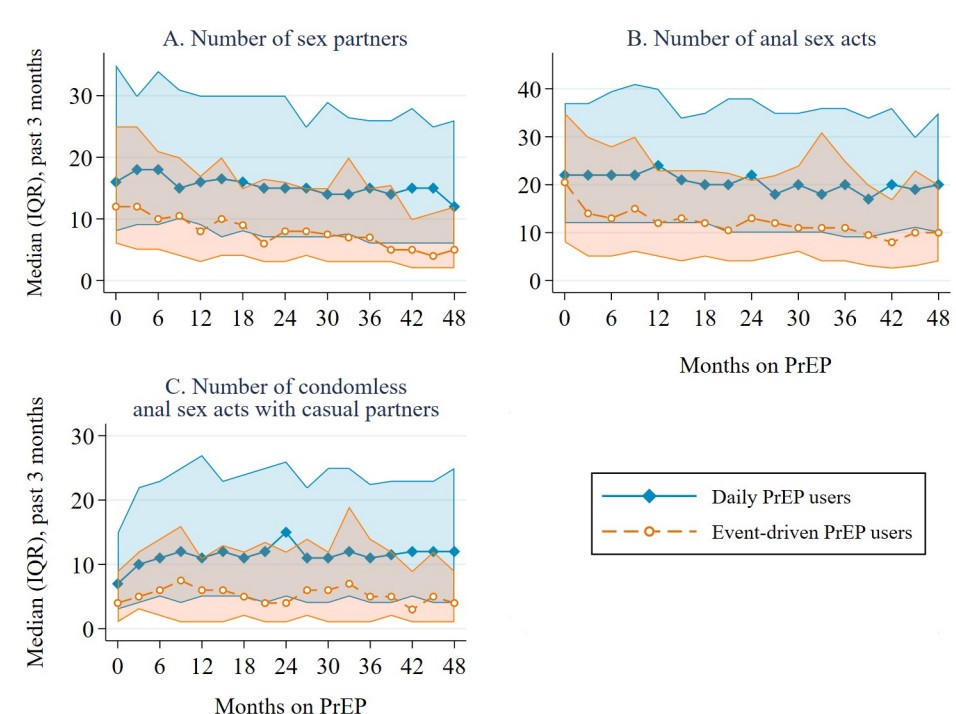

**Fig 1. Sexual behaviour of 367 AMPrEP participants during the first 4 years on PrEP, AMPrEP, the Netherlands, 2015–20.** Panel A shows median (IQR) number of sex partners, panel B shows median number of anal sex acts, and panel C shows median number of CAS acts with casual partners in the past 3 months. Lines represent medians per study visit. Shaded regions represent interquartile ranges. AMPrEP, Amsterdam PrEP demonstration project; CAS, condomless anal sex; IQR, interquartile range; PrEP, pre-exposure prophylaxis.

**Table 3. Four-year outcomes of incidence of STIs overall and by PrEP regimen, among 367 AMPrEP participants, Amsterdam, the Netherlands, 2015–2020.**

| | Total | | | | Daily PrEP | | | | Event-driven PrEP | | | | |
|---|---|---|---|---|---|---|---|---|---|---|---|---|---|
| | No. of participants with ≥1 positive test | No. of positive tests | PY | IR per 100PY [95% CI] | No. of participants with ≥1 positive test | No. of positive tests | PY | IR per 100PY [95% CI] | No. of participants with ≥1 positive test | No. of positive tests | PY | IR per 100PY [95% CI] | p-Value[a] |
| Any STI[b] | 289 | 1092 | 1,258 | 86.8 [81.8–92.1] | 246 | 891 | 914 | 97.4 [91.3–104.0] | 89 | 201 | 344 | 58.5 [50.9–67.2] | <0.0001 |
| Any anal STI[c] | 241 | 758 | 1,258 | 60.3 [56.1–64.7] | 208 | 630 | 914 | 68.9 [63.7–74.5] | 66 | 128 | 344 | 37.2 [31.3–44.3] | <0.0001 |
| Chlamydia[d] | 227 | 524 | 1,258 | 41.7 [38.2–45.4] | 197 | 432 | 914 | 47.2 [43.0–51.9] | 54 | 92 | 344 | 26.8 [21.8–32.8] | <0.0001 |
| Gonorrhoea[d] | 234 | 615 | 1,258 | 48.9 [45.2–52.9] | 195 | 505 | 914 | 55.2 [50.6–60.3] | 61 | 110 | 344 | 32.0 [26.6–38.6] | <0.0001 |
| Infectious syphilis[e] | 111 | 140 | 1,258 | 11.1 [9.4–13.1] | 86 | 108 | 914 | 11.8 [9.8–14.3] | 30 | 32 | 344 | 9.3 [6.6–13.2] | 0.27 |
| HIV | 2 | 2 | 1,258 | 0.2 [0.0–0.6] | 2 | 2 | 914 | 0.2 [0.1–0.9] | 0 | 0 | 344 | 0.0 [0.0–1.1] | 0.53 |
| HCV[f,g] | 15 | 17 | 1,186 | 1.4 [0.9–2.3] | 14 | 15 | 870 | 1.7 [1.0–2.9] | 2 | 2 | 315 | 0.6 [0.2–2.5] | 0.17 |

[a]Two-sided p-values for the crude incidence rate difference between daily and event-driven PrEP users were based on the Z-test.

[b]Any STI: chlamydia (any location), gonorrhoea (any location), infectious syphilis (stage 1, 2, and recent latent infection).

[c]Any anal STI: anal chlamydia or anal gonorrhoea.

[d]At any anatomical location.

[e]Syphilis stage 1, stage 2, and recent latent infection.

[f]Based on ribonucleic acid (RNA) positivity and regardless of history of prior HCV infection of HCV antibody positivity.

[g]One individual had a first HCV infection and HCV re-infection during follow-up.

AMPrEP, Amsterdam PrEP demonstration project; CI, confidence interval; HCV, hepatitis C virus; HIV, human immunodeficiency virus; IR, incidence rate; PrEP, pre-exposure prophylaxis; PY, person-years; STI, sexually transmitted infection.

## Regimen switching

Overall, 137/367 (37%) participants switched PrEP regimens 254 times: 141 times from daily PrEP to event-driven PrEP and 113 times from event-driven PrEP to daily PrEP use. The rate of any switch was highest in the first year (26.1/100PY) and decreased over time (IRR 0.87/year [95% CI 0.78–0.98], p = 0.019) to 17.2/100PY in the fourth year (S8 Table and Fig 4). The rate of switching from event-driven to daily PrEP was higher than vice versa, especially in the first year (S8 Table).

## Discontinuation of PrEP use

We observed 112 PrEP stops among 98/367 (27%) participants; 31 stops were subsequently followed by a restart. A total of 43 participants had a formal study exit visit and another 43 were lost to follow-up. We registered 17 gaps of more than 9 months between study visits, and 9 participants reported not having used PrEP for a period of at least 3 months, despite continuing study participation. Median time until the first stop among those who stopped was 21 (IQR = 12–35) months. Overall rate of stopping was 8.3/100PY (95% CI = 6.9–10.0). In multivariable analysis, younger age (p = 0.036), fewer CAS acts with casual partners (p = 0.039), not having a university/university of applied sciences degree (p = 0.030) and an MHI-5 score of <60 (p = 0.036) were associated with earlier stopping (S9 Table). Being diagnosed with an STI in the past 3 months was not associated with stopping (adjusted HR = 0.48, 95% CI = 0.14–1.64, p = 0.24; S9 Table).

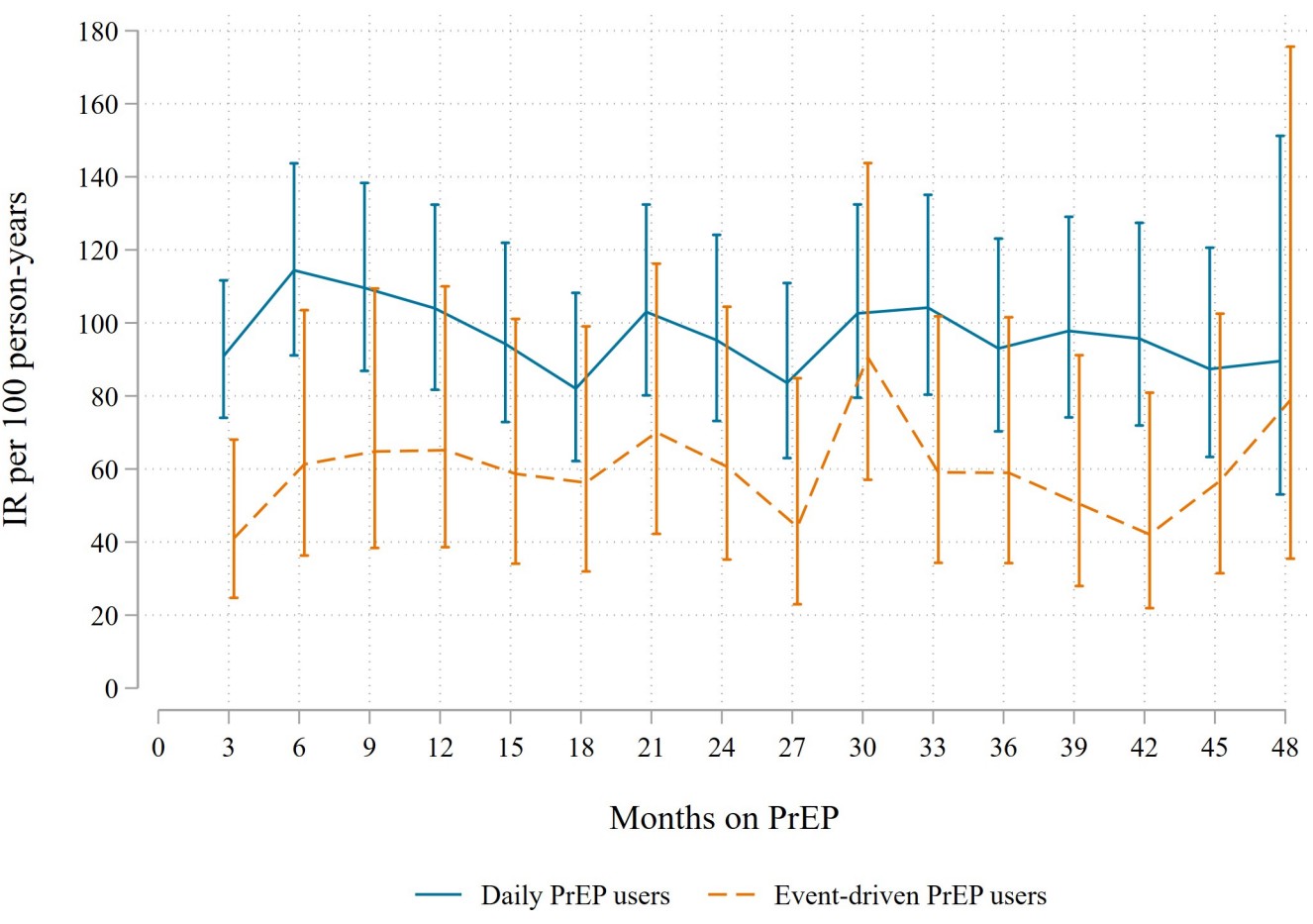

**Fig 2. Incidence of any syphilis, gonorrhoea, or chlamydia among 367 AMPrEP participants during the first 4 years on PrEP, AMPrEP, the Netherlands, 2015–2020.** Lines represent incidence rates per 100 person-years, as calculated over the previous 3 months. Vertical bars represent 95% CIs. AMPrEP, Amsterdam PrEP demonstration project; CI, confidence interval; IR, incidence rate; PrEP, pre-exposure prophylaxis.

### Intracellular TFV-DP concentrations

Among daily PrEP users, median TFV-DP concentration at 3 or 6 months was 1,263 fmol/punch (IQR = 1,000–1,619; $n$ = 240), at 12 months 1,299 fmol/punch (IQR = 1,021–1,627; $n$ = 259), at 24 months 1,288 fmol/punch (IQR = 1,005–1,617; $n$ = 223), and at 48 months 1,693 fmol/punch (IQR = 1,310–2,252; $n$ = 127). At 12 months, 93% ($n$ = 240/259) of participants had a TFV-DP concentration ≥700 fmol/punch, at 24 months 90% ($n$ = 200/223), and at 48 months 94% ($n$ = 120/127).

### Discussion

Over the first 4 years of PrEP use among participants of this prospective demonstration cohort in Amsterdam, the Netherlands, the number of CAS acts with casual partners and the total number of sex partners decreased over time. STI incidence was high, but stable over time. Therefore, these findings do not confirm apprehensions of increasing STIs in the first 4 years following PrEP initiation. Incidence of HIV was very low and the 2 incident infections occurred during the first year on PrEP. Objectively measured adherence among daily PrEP users remained well above the protective threshold during study follow-up for the large majority of participants. Retention at 48 months was high (68%). Thus,

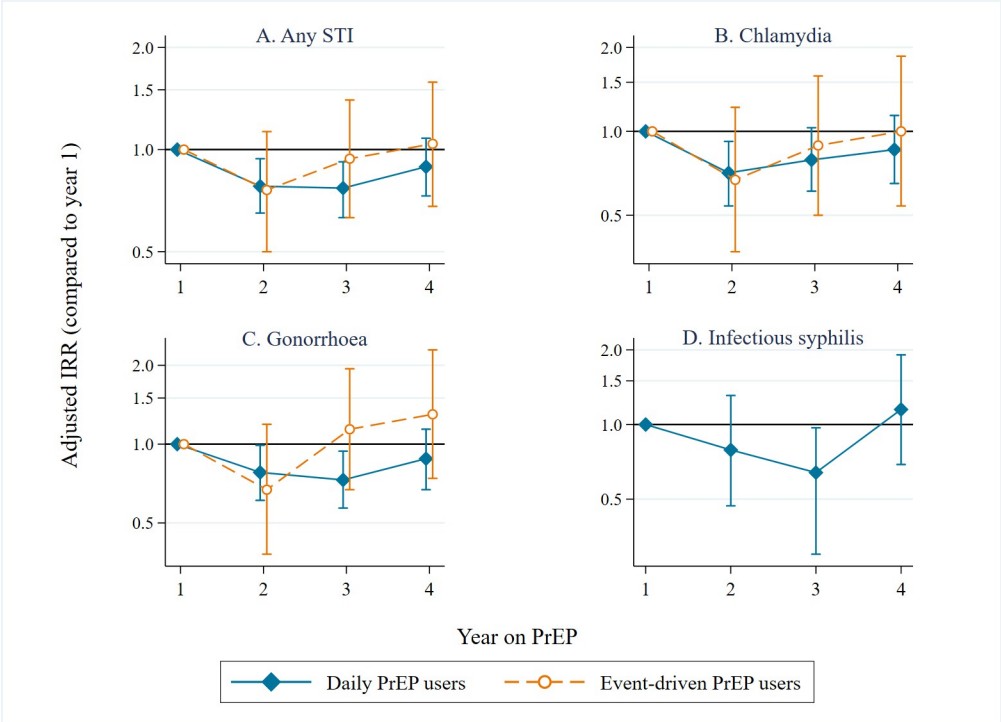

**Fig 3.** Incidence rate ratios per year on PrEP for any STI (A), chlamydia (B), gonorrhoea (C), and infectious syphilis (stage 1 and 2 and recent latent; D), adjusted for age and STI testing frequency, among 367 AMPrEP participants, Amsterdam, the Netherlands, 2015–2020. AMPrEP, Amsterdam PrEP demonstration project; IRR, incidence rate ratio; STI, sexually transmitted infection; NB, Year 1 is used as reference category; IRRs are adjusted for age at baseline and individual yearly STI testing frequency, modelled as restricted cubic splines with 4 knots. Due to a low number of newly detected infectious syphilis among event-driven PrEP users, we estimated the IRR per year on PrEP only for daily PrEP users. Vertical bars represent 95% CIs.

effectiveness of PrEP continues beyond the previously reported 2-year results in our and other demonstration studies [8,10,28].

PrEP programme policies select for people who are behaviourally susceptible for HIV and therefore these people are prone to acquire other STIs. As AMPrEP was initiated in 2015, before the European Medicine Agency approved TDF/FTC for PrEP in July 2016, there was no other formal way to acquire PrEP in the Netherlands at the time. The lack in PrEP availability likely resulted in inclusion of a group of early PrEP adopters. After generic PrEP became available in the Netherlands in 2017, a limited but growing number of general practitioners (GPs) started prescribing PrEP. The Dutch national PrEP pilot, offering PrEP at a reduced price and free PrEP-care at public health services to a maximum of 8,500 people, started in July 2019 [29]. AMPrEP participants were allowed to exit the study and enter the national PrEP pilot whenever they preferred to. Alternatively, they could be referred to their GP, but PrEP care through GPs in the Netherlands remains insufficiently accessible [30].

We assessed multiple sexual behaviour measures. We considered the number of CAS acts with casual partners as the most relevant factor in the context of HIV acquisition. The rate of this behaviour decreased over 4 years of follow-up in both daily PrEP and event-driven PrEP users, with expectedly higher numbers among daily PrEP users. The total number of sex partners also decreased over time, as previously observed by Molina and colleagues [8]. Grant and colleagues noted, in a double blind, randomised, placebo-controlled trial, a reduction of sex partners with whom participants had receptive intercourse over a median follow-up of 1.2

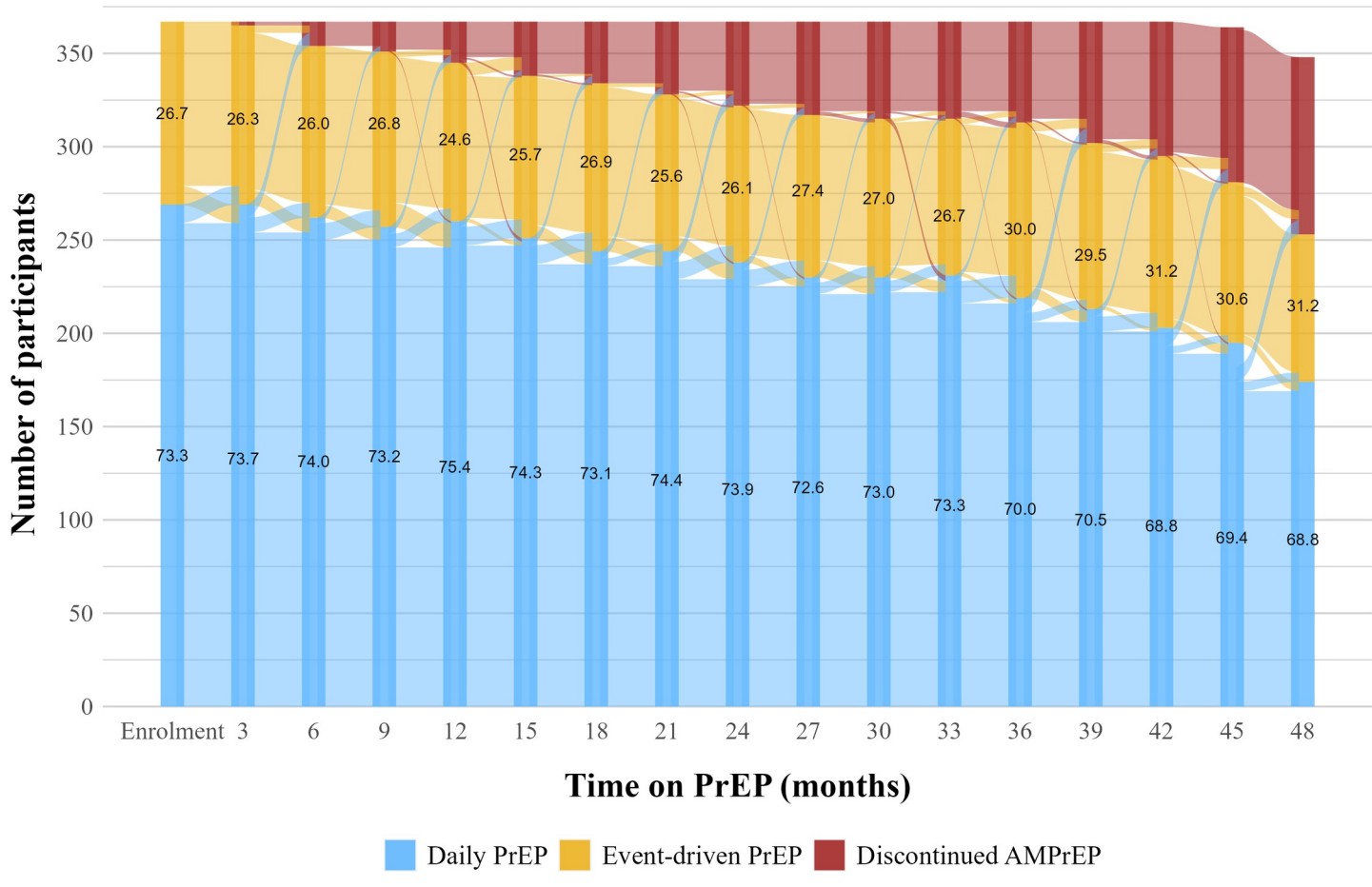

**Fig 4. Transitions through PrEP regimens during up to 4 years on PrEP among AMPrEP participants, Amsterdam, the Netherlands, 2015–2020.** At baseline, numbers on the bars indicate the percentage of participants who intended to use daily or event-driven PrEP. At all follow-up visits, numbers on the bars indicate the percentage of participants who reported having used daily or event-driven PrEP in the past 3 months. AMPrEP, Amsterdam PrEP demonstration project; PrEP, pre-exposure prophylaxis.

years, and suggested that the services around PrEP use (e.g., counselling) or taking the pill itself could serve as a reminder of HIV risk and contribute to choosing this "safer behaviour" [1]. Reyniers and colleagues suggest a broader paradigm of improved sexual health brought about by PrEP, through empowering its users to more actively engage in their sexual health [31]. AMPrEP participants had the opportunity to reflect on their sexual activity with nurses and physicians during each three-monthly consultation. This paradigm of sexual empowerment, leading to more considered sexual decisions, could explain why we observed reduced numbers of sex partners and CAS acts with casual partners over time in our cohort. An alternative explanation could be that sexual behaviour changes over time and that participants enrolled in this study when their need for PrEP was particularly high.

AMPrEP and the Be-PrEP-ared project in Belgium were, to the best of our knowledge, the first prospective demonstration projects to offer participants the choice between daily PrEP and event-driven PrEP, including the option to switch between regimens. We observed a high and stable incidence of STIs, similar to other early PrEP studies [32–34]. We noted this especially among daily PrEP users, as well as higher numbers of sex partners and CAS acts compared to event-driven PrEP users, in agreement with an earlier, pooled analyses of AMPrEP and Be-PrEP-ared over the first 28 months [35]. Daily PrEP appears to be used during periods

with more frequent sexual contacts or less condom use, coinciding with an increase in the chance to acquire HIV and STIs. This suggests that PrEP users are capable of deciding which regimen to use, depending on their sexual behaviour.

The low HIV incidence is likely a direct result of high PrEP adherence, as demonstrated by high median levels of TFV-DP around the level of perfect adherence. These levels were also well above the protective thresholds at all time points up to 4 years after PrEP initiation in the majority of participants. A subgroup of participants was included in a nested RCT assessing the effect of an app providing visualised feedback to increase adherence, as reported previously [24,36], which could have possibly been a contributing factor to adherence. The absence of any HIV infection after 1 year on PrEP provides further evidence that PrEP use can be sustainable and efficacious in preventing HIV over the longer course. This finding is in line with other studies with less follow-up time [8,28]. The stable, high STI rates are comparable to those in contemporaneous studies ranging from 75 to 98 per 100PY [34,37–39], although differences in methods could explain some of the variation between estimates. Post-exposure prophylaxis using doxycycline can effectively prevent bacterial STIs [40], and could, in the future, be considered for specific PrEP users. This is currently being adopted in some countries (e.g., the United States [41]), but not in the Netherlands, and long-term effects on antimicrobial resistance remain unknown [42,43].

HCV prevalence was high among participants initiating PrEP in AMPrEP [12] as described in previous analyses of our cohort, but during follow-up its incidence decreased over time, parallel to the decrease seen in the Dutch population with HIV since direct-acting antivirals for HCV became widely available in 2015 [44].

In our cohort, a substantial proportion of participants switched PrEP regimens once or multiple times, especially during the first year of PrEP use. The rate of switching from event-driven to daily PrEP was higher than vice versa. Previous qualitative research suggested recurring side-effects at re-initiation of PrEP during event-driven use, and difficulties in adhering to an irregular regimen, as reasons for this shift [45]. Retention to PrEP remained high over 4 years and the vast majority of the daily PrEP users had high adherence levels at each measurement. The majority of participants that stopped using PrEP did so because of low self-perceived need for PrEP, as reported previously [25,45]. However, stopping was also associated with younger age, not having a university/university of applied sciences degree and with having signs of depression/anxiety in our study. Furthermore, there can be discordance between self-perceived and actual need for PrEP [46]. This can leave ex-PrEP users vulnerable to HIV, as supported by reports of high HIV incidence among people who discontinued PrEP [47], stressing the importance of low-threshold access to PrEP and adherence and persistence counselling so PrEP users may discontinue PrEP well-informed. PrEP providers should make an effort to confirm that those who are lost to follow-up are using alternative HIV prevention strategies, or are invited into PrEP care again, if necessary.

A major strength of this study is the long follow-up time of up to 4 years. In real-world settings, many are likely to use PrEP for several years and very little data were available on long-term (i.e., longer than 2 years) PrEP use. Second, AMPrEP was a prospective, observational cohort, enabling a prospective assessment of outcome measures in which STI diagnoses made in-between study visits were included. Third, AMPrEP was among the first 2 demonstration projects allowing participants to choose between daily PrEP and event-driven PrEP use, adapt their PrEP use to match their need of protection, allowing an independent and long-term assessment of both regimens and inter-regimen switching behaviour.

We acknowledge some limitations. The AMPrEP cohort officially closed at 1 December 2020, but due to COVID-19 measures, we censored data after 15 March 2020, to exclude effects on sexual behaviour and STI incidence resulting from the COVID-19 pandemic.

Therefore, we were not able to include up to 5 years of follow-up nor the official end-of-study visits. Second, this cohort presumably included a high proportion of early adopters. Participants were relatively old, and mostly identified as men, white, and were university/university of applied sciences educated and they are unlikely to represent the broader population of MSM and transgender women who could benefit from PrEP. We were only able to include 2 transgender women.

This study's results support implementation of low-threshold PrEP services that make minimum demands on the user, have minimum criteria for access, and offer regular counselling, STI testing, and treatment. Future PrEP studies should aim to include a more representative sample for the population susceptible to HIV.

In conclusion, oral PrEP is a very effective HIV prevention strategy over a long period of use. We observed decreasing numbers of sex partners and CAS acts, and no increase in STI incidence over time. People susceptible to HIV should be empowered to choose their preferred HIV prevention strategies, including PrEP.

## Supporting information

**S1 Table. Sexual behaviour among 367 AMPrEP participants using daily versus event-driven PrEP over 4 years in Amsterdam, the Netherlands, 2015–2020.**
(DOCX)

**S2 Table. Four-year outcomes of sexual behaviour per year since PrEP initiation among 367 AMPrEP participants, Amsterdam, the Netherlands, 2015–2020.**
(DOCX)

**S3 Table. Four-year outcomes of incidence of STIs overall and by PrEP regimen, among 367 AMPrEP participants, Amsterdam, the Netherlands, 2015–2020.**
(DOCX)

**S4 Table. Four-year outcomes of incidence of STIs overall and by PrEP regimen, among 142 AMPrEP participants who ever switched their PrEP regimen, Amsterdam, the Netherlands, 2015–2020.**
(DOCX)

**S5 Table. Incidence rate ratio for STIs per additional year on PrEP over 4 years of PrEP use among 367 AMPrEP participants, Amsterdam, the Netherlands, 2015–2020.**
(DOCX)

**S6 Table. Incidence rate ratio for STIs per additional year since initiating PrEP (i.e., including gaps in PrEP use or AMPrEP participation) over 4 years among 367 AMPrEP participants, Amsterdam, the Netherlands, 2015–2020.**
(DOCX)

**S7 Table. Incidence rate ratio for STIs per additional year on PrEP over 4 years of PrEP use, among 225 AMPrEP participants who never switched PrEP regimens, Amsterdam, the Netherlands, 2015–2020.**
(DOCX)

**S8 Table. PrEP regimen switch rates during up to 4 years of PrEP use, among 367 AMPrEP participants, Amsterdam, the Netherlands, 2015–2020.**
(DOCX)

**S9 Table. Factors associated with an earlier time until first stopping PrEP among 367 AMPrEP participants, Amsterdam, the Netherlands, 2015–2020.**
(DOCX)

**S1 Annex. Acknowledgements of the H-TEAM consortium.**
(DOCX)

**S1 STROBE checklist. Statement—checklist of items that should be included in reports of observational studies.**
(DOCX)

## Acknowledgments

We thank all AMPrEP participants, members of the advisory board, and community engagement group. Additionally, we thank Roel Achterbergh, Ertan Ersan, Michelle Kroone, Dominique Loomans, Kees de Jong, Ilya Peters, Princella Felipa, Myra van Leeuwen, Jason Schouten, Kenneth Yap, Hanne Zimmermann, and all members of the H-TEAM consortium (S1 Annex).

## Author Contributions

**Conceptualization:** Mark A. M. van den Elshout, Udi Davidovich, Henry J. C. de Vries, Maria Prins, Maarten F. Schim van der Loeff, Elske Hoornenborg.

**Data curation:** Mark A. M. van den Elshout, Eline S. Wijstma, Liza Coyer.

**Formal analysis:** Mark A. M. van den Elshout, Eline S. Wijstma, Anders Boyd, Vita W. Jongen, Liza Coyer, Peter L. Anderson, Maarten F. Schim van der Loeff.

**Funding acquisition:** Maria Prins, Maarten F. Schim van der Loeff, Elske Hoornenborg.

**Investigation:** Mark A. M. van den Elshout, Elske Hoornenborg.

**Methodology:** Mark A. M. van den Elshout, Eline S. Wijstma, Anders Boyd, Vita W. Jongen, Maarten F. Schim van der Loeff.

**Project administration:** Mark A. M. van den Elshout.

**Resources:** Mark A. M. van den Elshout.

**Supervision:** Maria Prins, Maarten F. Schim van der Loeff, Elske Hoornenborg.

**Validation:** Mark A. M. van den Elshout, Eline S. Wijstma, Anders Boyd, Vita W. Jongen, Liza Coyer, Maarten F. Schim van der Loeff.

**Visualization:** Mark A. M. van den Elshout, Eline S. Wijstma.

**Writing – original draft:** Mark A. M. van den Elshout.

**Writing – review & editing:** Mark A. M. van den Elshout, Eline S. Wijstma, Anders Boyd, Vita W. Jongen, Liza Coyer, Peter L. Anderson, Udi Davidovich, Henry J. C. de Vries, Maria Prins, Maarten F. Schim van der Loeff, Elske Hoornenborg.

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
