## [Editor Report · Decision Letter 0]

7 Dec 2023

Dear Dr van den Elshout, 

Thank you for submitting your manuscript entitled "Four years of PrEP use; sexual behaviour and STIs in the AMPrEP demonstration project cohort among men who have sex with men in Amsterdam, the Netherlands" for consideration by PLOS Medicine.

Your manuscript has now been evaluated by the PLOS Medicine editorial staff and I am writing to let you know that we would like to send your submission out for external peer review.

Please re-submit your manuscript within two working days, i.e. by Dec 11 2023 11:59PM.

Feel free to email me at lgaynor@plos.org if you have any queries relating to your submission.

Kind regards,

Louise Gaynor-Brook, MBBS PhD

---

## [Decision Letter · Decision Letter 1]

17 Jan 2024

Dear Dr. van den Elshout,

Many thanks for submitting your manuscript "Four years of PrEP use; sexual behaviour and STIs in the AMPrEP demonstration project cohort among men who have sex with men in Amsterdam, the Netherlands" (PMEDICINE-D-23-03584R1) to PLOS Medicine. The paper has been reviewed by two subject experts and a statistician; their comments are included below and can also be accessed here: [LINK]

As you will see, the reviewers were very positive about the paper and the importance of these follow-up data, but they raised a number of questions about specific study details and presentation. After discussing the paper with the editorial team and an academic editor with relevant expertise, I’m pleased to invite you to revise the paper in response to the reviewers’ comments. We plan to send the revised paper to some of all of the original reviewers*, and of course we cannot provide any guarantees at this stage regarding publication. 

When you upload your revision, please include a point-by-point response that addresses all of the reviewer and editorial points, indicating the changes made in the manuscript and either an excerpt of the revised text or the location (eg: page and line number) where each change can be found. Please submit a clean version of the paper as the main article file and a version with changes marked should as a marked up manuscript. Please also check the guidelines for revised papers at http://journals.plos.org/plosmedicine/s/revising-your-manuscript for any that apply to your paper.

We ask that you submit your revision by Feb 7th. However, if this deadline is not feasible, please contact me by email, and we can discuss a suitable alternative. 

Don’t hesitate to contact me directly with any questions (hvanepps@plos.org). If you reply directly to this message, please be sure to ‘Reply All’ so your message comes directly to my inbox. 

Kind regards,

Heather

Heather Van Epps, PhD

Executive Editor

[on behalf of]

Louise Gaynor-Brook, MBBS PhD

PLOS Medicine

plosmedicine.org

*Please note: If your article is accepted, you may have the opportunity to make the peer review history publicly available. The record will include editor decision letters (with reviews) and your responses to reviewer comments. If eligible, we will contact you to opt in or out.

Editorial comments:

1. Because you have allowed participants to switch between daily PrEP and event-driving (intermittent) PrEP (the IR quoted are then based on person-years of observations on that regimen), individuals can contribute to both regimens for different periods of exposure time. This potentially makes interpretation of the data somewhat less intuitive. Might it be possible to explore more what happened within individuals when switching - eg what was incidence of STI in the daily PrEP versus incidence in event-driven PrEP in the same person? This might allow you to confirm the suggestion (in the discussion) that people adjust their regimen based on self-perceived need for PrEP.

2. Regarding point #4 raised by Reviewer 3 (“Is PrEP now available through GP in Amsterdam and thus, some participants may choose to seek a GP rather than continuing to have scheduled visits with blood sample drawn at each visit after a few years?”), we felt it might be useful to include some detail about the Dutch healthcare system to make it clear how these therapies are dispensed. It is my understanding that the Netherlands do indeed have general practitioners (although perhaps not referred to as ‘GPs’), but this differs in other health systems, so it would be useful to include some contextual detail in the paper around this point.

Comments from the reviewers:

Reviewer #1 (statistics): 

See attachment

Michael Dewey

Reviewer #2: 

This is a well-written manuscript describing 4 years of follow-up of the PrEP demonstration project conducted in Amsterdam. Other published PrEP data have substantially smaller numbers, do not include 2-1-1 PrEP, and/or have shorter follow-up periods, so this represents a substantial contribution to the published literature on PrEP. The important limitation that this was largely in older, white, well-educated men is commented on in the limitations section of the manuscript.

Major comments:

1. Although the manuscript is focused on "risk compensation," the value of the manuscript really resides in reporting on a large cohort of MSM taking PrEP. Risk compensation is concept that some would not feel is important to track, given the high effectiveness of HIV PrEP and the possibility now for the use of doxycycline post-exposure prophylaxis (Doxy-PEP).

2. It appears that the rate of switching from event-driven to daily was substantially higher than the converse, but that is not clear from the results section (which only capture the number who switched, without regard to the size of the two subgroups). This also is not commented upon in the discussion, which could strengthen the manuscript.

Minor comments:

1. It would be useful for the authors to cite the STI rates in comparable contemporaneous studies globally.

2. There is no mention of Doxy-PEP, despite the high STI rates. 

3. The authors collected data on sexual satisfaction, but that was not included in the manuscript. That would be of great interest, as this is another potential contribution of HIV PrEP to well-being.

4. It appears (from table S8) that mental health was also significantly associated with stopping PrEP, but this is not mentioned in the results or conclusions.

Reviewer #3: 

This is a very timely and important article with up to 4-year data on HIV incidence, sexual behavioral change and STI incidence in PrEP users in Amsterdam. This study shows that PrEP users are already at high-risk for bacterial and viral STI before starting PrEP, and that incidence rates do not rise while on PrEP, which does not suggest a risk compensation behavior as stated by many physicians and advocates who are fighting against PrEP. 

I have minor comments in order to improve the manuscript:

1. Line 129 : « We defined any STI as having one or more bacterial STIs (i.e., chlamydia, gonorrhoea or infectious syphilis) at a visit » : did you recommend a « test-of-cure » after the treatment of an STI to ensure that a new episode was really a re-infection rather than the same previous infection that was not (correctly) treated?

2. Line 209 : « The median follow-up time was 3.9 years (IQR=3.4-4.0) ». The title states that these are the 4-year results, but a significant proportion of individuals did not reach 4 years of follow-up. The title is thus misleading. I suggest that the « 4 years » be removed from the title for more consistency with the results presented herein.

Discussion:

3. Do you have data on condom use ? Were some of the CAS protected with PrEP and with condom? Were some of the CAS protected with condom with no PrEP use in the event-driven users?

4. Do you have some thoughts about lost-to-follow-up ? are they real LTFU ? or did they move to another city and were still taking PrEP? Is PrEP now available through GP in Amsterdam and thus, some participants may choose to seek a GP rather than continuing to have scheduled visits with blood sample drawn at each visit after a few years? This should be discussed more thoroughly.

5. Line 356 : « However, there can be discordance between self-perceived and actual need for PrEP » : I totally agree with this statement, and I think some targeted interventions are needed for those who discontinue PrEP in order to make sure that they do not actually need it. This should be discussed more thoroughly.

6. Line 379 : « implementation of low-threshold PrEP services » : can you explain what is meant by « low thershold PrEP services »?

[LINK]

1. Please upload any figures associated with your paper as individual TIF or EPS files with 300dpi resolution at resubmission; please read our figure guidelines for more information on our requirements: http://journals.plos.org/plosmedicine/s/figures. While revising your submission, please upload your figure files to the PACE digital diagnostic tool, https://pacev2.apexcovantage.com/. PACE helps ensure that figures meet PLOS requirements. To use PACE, you must first register as a user. Then, login and navigate to the UPLOAD tab, where you will find detailed instructions on how to use the tool. If you encounter any issues or have any questions when using PACE, please email us at PLOSMedicine@plos.org.

To submit your revised manuscript: 

---

## [Decision Letter · Decision Letter 2]

16 Mar 2024

Dear Dr. van den Elshout,

Thank you very much for re-submitting your manuscript "Four years of PrEP use; sexual behaviour and STIs in the AMPrEP demonstration project cohort among men who have sex with men in Amsterdam, the Netherlands" (PMEDICINE-D-23-03584R2) for review by PLOS Medicine.

I have discussed the paper with my colleagues and the academic editor and it was also seen again by one reviewer. I am pleased to say that provided the remaining editorial and production issues are dealt with we are planning to accept the paper for publication in the journal.

[LINK]

If you have any questions in the meantime, please contact me on lgaynor@plos.org.  

We look forward to receiving the revised manuscript by Mar 25 2024 11:59PM.   

Sincerely,

Louise Gaynor-Brook, MBBS PhD

Senior Editor 

PLOS Medicine

plosmedicine.org

Requests from Editors:

Thank you for your patience with a longer assessment process than we anticipated, and apologies for the delay in providing you with an editorial decision. The list below appears rather lengthy, but some of these points are more minor points which should not require a substantial amount of time to attend to. 

General comments:

Throughout the paper, please adapt reference call-outs to the following style: "... every year [1,2]." (noting the absence of spaces within the square brackets).

To help us extend the reach of your research, please provide any Twitter handle(s) that would be appropriate to tag, including your own, your coauthors’, your institution, funder, or lab.

Data availability:

Please note that a study author cannot be the contact person for data requests. Please confirm that the email address provided (amprep@ggd.amsterdam.nl) is independent of the study authors. 

Title: Please revise your title according to PLOS Medicine's style. Please place the study design in the subtitle (ie, after a colon). We suggest “Sexual behaviour and incidence of sexually transmitted infections among men who have sex with men (MSM) and transgender women using daily and event-driven pre-exposure prophylaxis: Four-year follow-up of the Amsterdam Pre-Exposure Prophylaxis (AMPrEP) demonstration project cohort” or similar 

Abstract:

Please structure your abstract using the PLOS Medicine headings (Background, Methods and Findings, Conclusions), combining the Methods and Findings sections into one section.

Please define STI at first use.

Abstract Methods and Findings:

Please ensure that all numbers presented in the abstract are present and identical to numbers presented in the main manuscript text.

Please provide brief demographic details of the study population (e.g. sex, age, ethnicity, etc)

Please include the study design, and the years during which follow-up took place. 

Please define IQR at first use.

Please include the important dependent variables that are adjusted for in the analyses.

Please define CI at first use.

Please include the actual numbers of relevant outcomes. 

Please include a summary of adverse events if these were assessed in the study.

In the last sentence of the Abstract Methods and Findings section, please describe 2-3 of the main limitations of the study's methodology.

Abstract Conclusions:

Please begin your Abstract Conclusions with "In this study, we observed ..." or similar, to summarize the main findings from your study, without overstating your conclusions. Please emphasize what is new and address the implications of your study, being careful to avoid assertions of primacy. 

Please remove the subsection on funding.

Author Summary:

In the final bullet point of ‘What Do These Findings Mean?’, please describe the main limitations of the study in non-technical language.

Introduction:

Line 72 - please define STI at first use.

Methods:

Please refer to your prospective early in the Methods section, and please indicate if/when reported analyses differed from those that were planned. Changes in the analysis-- including those made in response to peer review comments-- should be identified as such in the Methods section of the paper, with rationale. If a reported analysis was performed based on an interesting but unanticipated pattern in the data, please be clear that the analysis was data-driven.

Please include your Ethics statement in the Methods section.

Please add the following statement, or similar, to the Methods: "This study is reported as per the Strengthening the Reporting of Observational Studies in Epidemiology (STROBE) guideline (S1 Checklist)." The STROBE guideline can be found here: http://www.equator-network.org/reporting-guidelines/strobe/ When completing the checklist, please use section and paragraph numbers, rather than page numbers which will no longer correspond to the appropriate sections after copy-editing.

Results: 

Please include a table showing the baseline characteristics of the study population as Table 1 in the main manuscript. 

Please ensure that all numbers presented in the main text are identical to numbers presented in tables e.g. line 222 - aRR of CAS acts with casual partners does not appear to correspond with Table 1. Line 271 - IR for diagnosis of HIV during follow-up and Table 2/S4. 

Please define the length of follow up (eg, in mean, SD, and range).

Where aRR are presented, please specify the comparison group and indicate which factors are adjusted for. 

Line 245 - “STI incidence did not vary between 3-monthly periods within each year of PrEP use” - please indicate where data are shown. 

Line 271 - please refer to Table 2 where these data are also shown. 

Please report your results to the same number of decimal places in both the tables and manuscript text.

Line 294 - “Discontinuation of PrEP use” - please indicate where data are shown. 

Line 305 - “Intracellular TFV-DP concentrations” - please indicate where data are shown. 

Discussion:

Please present and organize the Discussion as follows: a short, clear summary of the article's findings; what the study adds to existing research and where and why the results may differ from previous research; strengths and limitations of the study; implications and next steps for research, clinical practice, and/or public policy; one-paragraph conclusion.

Lines 348, 394 - please temper assertions of primacy by adding ‘to the best of our knowledge’ or similar.

Please remove all subheadings within your Discussion e.g. Limitations and other considerations

Please remove the information on competing interests, funding and data sharing from the

end of the main text. In the event of publication, this information will appear in the article

metadata, via entries in the submission form.

Tables:

When a p value is given, please specify the statistical test used to determine it in the table legend of each table (including in the supplementary material).

Please report your results to the same number of decimal places in both the tables and manuscript text.

Please define all abbreviations used in the table legend of each table (including in the supplementary material).

References:

Please ensure that journal name abbreviations match those found in the National Center for Biotechnology Information (NCBI) databases (http://www.ncbi.nlm.nih.gov/nlmcatalog/journals), and are appropriately formatted and capitalised.

Please also see https://journals.plos.org/plosmedicine/s/submission-guidelines#loc-references for further details on reference formatting. 

Where website addresses are cited, please specify the date of access. 

Comments from Reviewers:

Reviewer #1: The authors have addressed all my points.

Michael Dewey

[LINK]

---

## [Editor Report · Decision Letter 3]

10 Apr 2024

Dear Dr van den Elshout, 

On behalf of my colleagues and the Academic Editor, Prof. Marie-Louise Newell, I am pleased to inform you that we have agreed to publish your manuscript "Sexual behaviour and incidence of sexually transmitted infections among men who have sex with men (MSM) using daily and event-driven pre-exposure prophylaxis (PrEP): Four-year follow-up of the Amsterdam PrEP (AMPrEP) demonstration project cohort" (PMEDICINE-D-23-03584R3) in PLOS Medicine.

PRESS

Sincerely, 

Louise Gaynor-Brook, MBBS PhD 

Senior Editor 

PLOS Medicine